# Endophytic and epiphytic metabarcoding reveals fungal communities on cashew phyllosphere in Kenya

**Dennis Wamalabe Mukhebi**[1], **Colletah Rhoda Musangi**[1], **Everlyne Moraa Isoe**[1], **Johnstone Omukhulu Neondo**[2], **Wilton Mwema Mbinda**[1,3]*

**1** Department of Biochemistry and Biotechnology, Pwani University, Kilifi, Kenya, **2** Institute for Biotechnology Research, Jomo Kenyatta University of Agriculture and Technology, Juja, Kenya, **3** Pwani University Biosciences Research Centre (PUBReC), Pwani University, Kilifi, Kenya

\* wilton.mbinda@gmail.com, w.mbinda@pu.ac.ke

**Data Availability Statement:** The datasets generated and analyzed during the current study are available in the NCBI's Short Read Archive (SRA) under the Sequence Read Archive (SRA)

## Abstract

Plants intimately coexist with diverse taxonomically structured microbial communities that influence host health and productivity. The coexistence of plant microbes in the phyllosphere benefits biodiversity maintenance, ecosystem function, and community stability. However, differences in community composition and network structures of phyllosphere epiphytic and endophytic fungi are widely unknown. Using Illumina Miseq sequencing of internal transcribed spacer (ITS) and 28S rRNA gene amplicons, we characterised the epiphytic and endophytic fungal communities associated with cashew phyllosphere (leaf, flower and fruit) from Kwale, Kilifi and Lamu counties in Kenya. The ITS and 28S rRNA gene sequences were clustered into 267 and 108 operational taxonomic units (OTUs) at 97% sequence similarity for both the epiphytes and endophytes. Phylum *Ascomycota* was abundant followed by *Basidiomycota*, while class *Saccharomycetes* was most dominant followed by *Dothideomycetes*. The major non-ascomycete fungi were associated only with class *Tremellales*. The fungal communities detected had notable ecological functions as saprotrophs and pathotrophs in class *Saccharomyectes* and *Dothideomycetes*. The community composition of epiphytic and endophytic fungi significantly differed between the phyllosphere organs which was statistically confirmed by the Analysis of Similarity test (ANOSIM Statistic R: 0.3273, for 28S rRNA gene and ANOSIM Statistic R: 0.3034 for ITS). The network analysis revealed that epiphytic and endophytic structures were more specialized, modular and had less connectance. Our results comprehensively describe the phyllosphere cashew-associated fungal community and serve as a foundation for understanding the host-specific microbial community structures among cashew trees.

## Introduction

In natural ecosystems, plants intimately coexist with diverse taxonomically structured communities of microorganisms, collectively known as the microbiota. Plant microbiota comprises of

BioProject ID: PRJNA1051575, http://www.ncbi.nlm.nih.gov/bioproject/1051575.

**Funding:** This research was supported by the National Research Fund, Kenya (NRF/2/MMC/158) and The World Academy of Sciences (RGA No. 21-302 RG/BIO/AF/AC_G). The funders had no role in the design of the study; in the collection, analyses, or interpretation of data; in the writing of the manuscript, or in the decision to publish the results.

**Competing interests:** The authors have declared that no competing interests exist.

a diverse and rich community of bacteria, fungi, viruses, cyanobacteria, actinobacteria, nematodes, and protozoans [1, 2]. The microbiome (microbiota and their genomes) inhabiting the rhizosphere (below ground), endosphere (internal tissue matrix), phyllosphere (above ground) and other plant tissues establish complex and dynamic interactions with the host plant [3]. Since the plant microbiome covers a varied functional gene pool from either the plant rhizosphere or phyllosphere, it has drawn much attention due to its extensive microbial composition influencing plant growth and health [4].

Plant microbial composition and diversity are strongly influenced by the environmental factors such as temperature fluctuations, ultraviolet radiations, relative humidity, and desiccation [1]. Additionally, different parts of the phyllosphere influence microbial abundance and diversity. For instance, plant leaves with thin waxy cuticle layer around the upper and lower epidermis offer a physical barrier against various biotic and abiotic stressors [5]. Plant rhizosphere, phyllosphere, and endosphere are the primary causes of variations in the microbiome. Epiphytes and endophyte diverse within and among plant rhizosphere and phyllosphere microbiomes due to ecological nutritional differences [6]. Climate change and anthropogenic factors also affect microbial diversity within these microbiomes [7]. As rates of environmental change intensify, alterations in temporal dynamics of plant microbial functioning and biogeochemical cycling are expected to occur, hence the need to understand their impacts on plant growth and health.

The phyllosphere relies on the plant organs' nutritional composition, texture and inner matrix to elucidate microbial diversity and variations in the interactions among epiphytes and endophytes [8]. This nutrient-poor niche is a significant ecological unit in biodiversity maintenance, community composition and ecosystem function [6]. The epiphytic and endophytic fungal communities exhibit high species diversity variations among them due to the different microenvironments they occupy [6]. The epiphytic fungi are in direct contact with the external environment, and utilize nutrient deposits on leaves, flowers, fruits and stems acquired from the atmosphere or plant exudates, unlike endophytic fungi that are resident in tissue matrix and use the host nutrients [9]. Fungal endophytes confer biotic and abiotic stress resistance, act as biofertilizers [1], phytostimulators [2] and biopesticides [10]. Endophytic fungi are horizontally transmitted and exploit the phyllosphere surfaces more often by circumventing the plant immune system [6]). Studies among strawberries [11] and tropical mangrove forests [12] have unravelled the phyllosphere microbial composition and their role in ecosystem functioning. Plant phyllosphere fungal composition for each tissue matrix (leaf, flower and fruit) differ [13]. Additionally, epiphytic and endophytic communities synergistically exhibit mutualistic effects in litter decomposition, thereby promoting carbon and nutrient recycling which is necessary for plant health and growth [6]. Symbiotic and pathogenic fungal communities coexist within the microenvironments of plant phyllosphere, triggering intricate plant-microbe interactions [14]. This makes it imperative to understand the plant phyllosphere microbial composition of the cashew phyllosphere.

Cashew (*Anacardium occidentale*) is native to Central and South America, with eastern Brazil as the core diversity epicentre [15]. The Portuguese introduced cashew crops in Africa in the second half of the 16[th] Century [16] as an extensive forest cover for soil architecture maintenance, afforestation, and later an economically viable cash crop [17]. Cashew is globally grown for its nuts, apples, gum, and wood, making it an important source of income for small-scale farmers and the national economy [18]. In Kenya, cashew trees are predominantly grown in along the Indian ocean coastal belt [19].

Decline in cashew production across agro-ecological zones is linked to extensive monoculture cropping [20], human activities, climate change and plant pathogens [1]. The advent of molecular tools and their application in environmental genomics and metagenomics analyses

has revolutionized studies in microbiome composition and diversity characterization [3]. The amplicon metagenomics approach is key in understanding complex and dynamic plant microbiomes [10]. In this study, we used amplicon metagenomics approach to profile the epiphytic and endophytic fungal communities associated with cashew phyllosphere microbiome from coastal Kenya.

## Materials and methods

### Sample collection

This research was conducted among farmers from three counties in conjunction with relevant agricultural authorities through the expertise of the County Agricultural Extension Officers. The cashew phyllosphere was collected from the fields using non-lethal collection methods and shipped to the laboratory for further processing. The cashew phyllosphere sample materials (leaf, flower and fruit) were collected from Kwale, Kilifi and Lamu counties in coastal Kenya using a complete randomized stratified approach [21, 22]. Farms were grouped into 27 strata and 75 samples were collected: 21 in Kwale, 42 in Kilifi and 12 in Lamu. Identification of sampling points from each County was based on the size and availability of cashew trees. Collected samples were kept in falcon tubes containing 20% dimethyl sulfoxide, ethylenediaminetetraacetic acid, saturated salt buffer [23] and transported to Pwani University Bioscience Research Centre (PUBReC), Pwani University, Kilifi, Kenya, in a thermally insulated box at 4 ºC. The sample materials were stored at -80 ºC for further analysis. All experiments on cashew phyllosphere were carried out in accordance with relevant PUBReC standard operating procedures and regulations.

### Collection of fungal epiphytes and endophytes

The storage buffer and individual sample materials were emptied into a 100 mL conical flask and transferred to a rotary shaker for two hours at 180 revolutions per minute to allow the dislodgment of fungal spores for epiphyte collection. The storage buffer was then filtered through a sterile filter membrane (0.2 μm) fitted on a micro funnel to capture the fungal spores displaced from the tissue. Captured fungal spores on the sterile filter membrane were transferred into a 2 mL microcentrifuge tube using sterile forceps and deep-frozen in liquid nitrogen. The plant organs were rinsed thrice with double distilled water to remove the buffer. Surface sterilization of the organs for endophyte isolation was conducted sequentially as follows: 1 min in sterile distilled water, 1 min in 75% v/v of ethanol, 3 min in 3.25% sodium hypochlorite, and 30 secs in 75% v/v ethanol [6]. The organs were then transferred into 2 mL microcentrifuge tubes and deep-frozen in liquid nitrogen. Sterilization efficiency was tested using the last rinse by plating independently three plates using potato dextrose agar (PDA) followed by incubation for three days at 28 ºC.

### Total fungal DNA extraction and sequencing

The genomic DNA (gDNA) was extracted from homogenized epiphytes and endophytes. The samples were crushed under liquid nitrogen using sterilized and chilled mortar and pestle in a laminar airflow to avoid cross and external contamination. The gDNA was isolated from 1 g of both epiphytes and endophytes using Qiagen DNeasy Mini kit (Qiagen, Maryland, USA) in accordance with the manufacturer's protocol and quantified using a Nanodrop 2000 spectrophotometer (Thermo Fisher Scientific, Massachusetts, USA). The DNA integrity was confirmed using 1% agarose gel electrophoresis. For maximum capture and retention of the environmental microbial communities, the gDNA of epiphytes and endophytes from similar

sampling sites (per county) were pooled together, yielding 18 samples. A set universal primer (ITS1 and ITS4 and NL1 and NL4) (S1 Table in S1 File) for ITS and 28S rRNA gene regions, respectively were used. The gDNA was sent for paired-end 2 × 300 bp sequencing on an Illumina MiSeq platform at Macrogen Inc. (Seoul, Republic of Korea).

## Data processing and statistical analysis

The raw fastq reads were processed using FastQC (http://www.bioinformatics.babraham.ac.uk/projects/fastqc/) for pre and post quality check and trimmed using Trimmomatic tool [24]. Sequence analyses were performed with VSEARCH [25]. The reads for ITS and 28S rRNA gene were merged and concatenated into a single file, respectively. Both merged ITS and 28S rRNA reads were filtered, dereplicated, denoised, chimera removed and clustered to produce operational taxonomic units (OTUs) at 97% similarity using the USEARCH UPARSE pipeline. Greedy clustering using *usearch_global* function was applied for read mapping and OTU table creation. The fasta file of delimited OTUs was aligned using MAFFT with 1000 maximum iteration [26]. A Newick tree was created using Fast Tree package [27]. The Newick tree was annotated in R using ggtree package [28] for inferring phylogenetic relationships within the fungal microbial communities with both barcode genes. BLASTn local database was created using the Makeblastdb function for sequences from UNITE [29] and LSU 28S rRNA gene (https://www.arb-silva.de/browser/lsu/) databases, generating indexed files with extensions.nhr,.nin, and.nsq. BLASTn was used in OTU classification and taxonomic assignment, generating a taxa file from the respective database. The files were cleaned using a Python script to remove the underscores. The OTU table was rarefied to an even depth to reduce the biases in sequencing depth. Shannon's diversity index [30] and Chao1 estimate were calculated to determine the alpha fungal diversity. The impact of distinct microenvironments on OTU richness was evaluated using the Kruskal—Wallis test [31] and one-way analysis of variance (ANOVA). The weighted and unweighted UniFrac distance matrices were applied for non-metric multidimensional scaling (NMDS) ordination. A two-dimensional plane was used to determine whether communities with similar characteristics tend to cluster together. The dissimilarity was confirmed using the Analysis of Similarity test. Functional prediction of the associated cashew phyllosphere fungal communities was carried using FUNGildR under GNU General Public License hosted by GitHub (https://github.com/UMNFuN/FUNGuild). The igraph and Hmisc packages were used to create fungal community network on a phyloseq object. The OTU table was extracted and subjected to a correlation matrix using the Spearman method by pairwise complete observation. A clustering coefficient is created after running the degree of distribution. Using the walktrap community algorithm, fungal community modularity was calculated from the ITS and 28S rRNA gene to understand the strength within and between modules.

## Results

### Sequence statistics

After quality evaluations for both ITS and D1D2 28S rRNA gene demultiplexed fastq sequence yielded 389456 and 299998 high-quality sequence reads. Quality reads were clustered into 267 (S2 Table in S2 File) and 108 (S3 Table in S3 File) OTUs with undetected singletons and doubletons. Of the 267 OTUs of ITS, 155 OTUs were epiphytes, and 122 OTUs were endophytes while the 108 OTUs of 28S rRNA gene, 68 OTUs were epiphytes, and 40 OTUS were endophytes (S2 Table in S2 File, S3 Table in S3 File). The sequence read analysis for the ITS and 28S rRNA gene are outlined and described in the legend (Fig 1). The rarefaction curves for ITS and 28S rRNA gene reached an asymptote, revealing a maximum detection of fungal species.

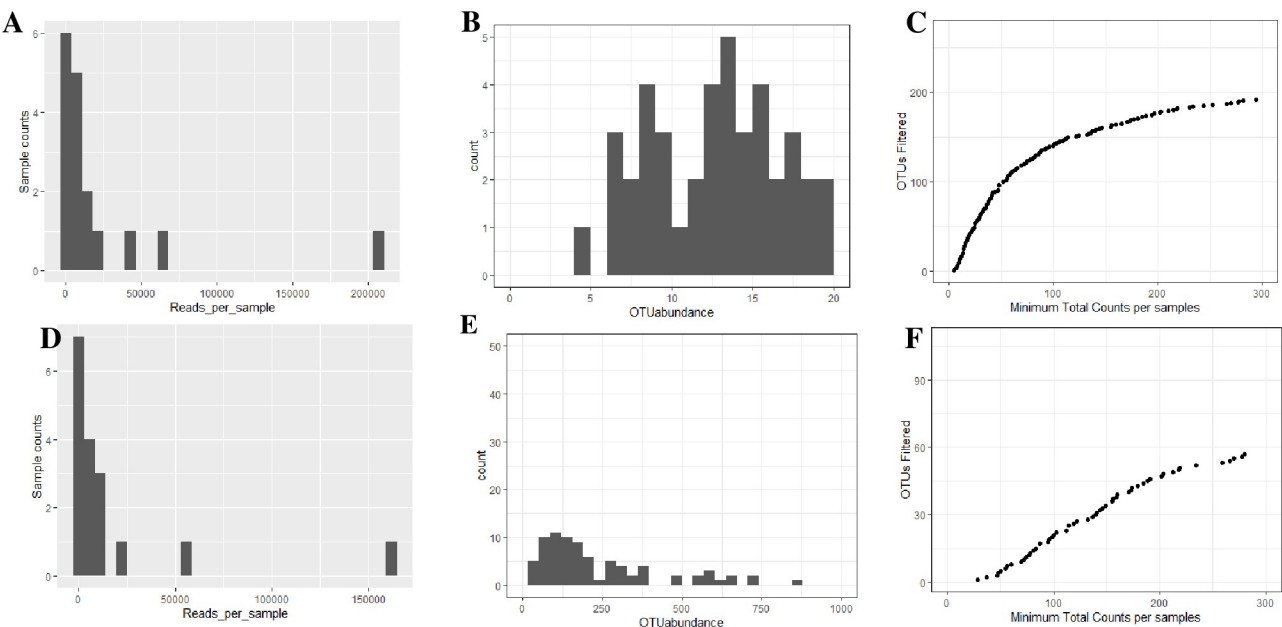

**Fig 1. Sequence analysis reads. A**. read distribution per sample for ITS, **B.** OTU abundance per sample for ITS, **C.** Rarefaction Curve for ITS reads, **D.** reads distribution per sample for 28S rRNA gene, **E.** OTU abundance per sample for 28S rRNA gene, **F.** Rarefaction Curve for 28S rRNA gene.

## Fungal diversity analysis of plant organs, epiphytes, and endophytes

The alpha diversity within the plant organs, epiphytes, and endophytes, as well as the study sites for both ITS and 28S rRNA gene (Fig 2), showed moderate species richness and evenness according to Shannon index values with no significant difference. The NMDS ordination

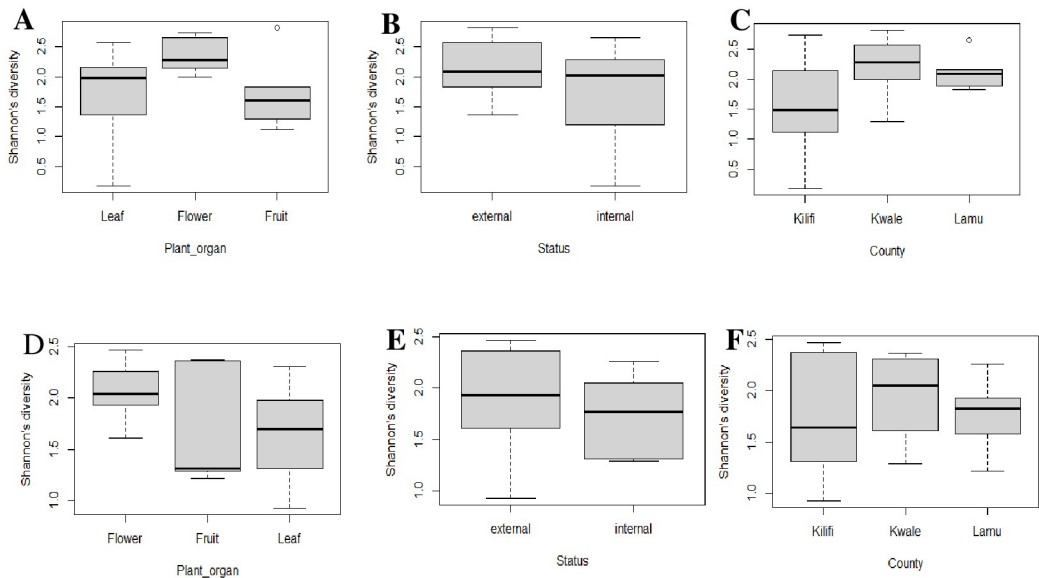

**Fig 2.** Box plot visualizing results of one-way ANOVA to comparing Shannon diversity: **A.** plant organs (p = 0.193), **B.** epiphytes and endophytes (p = 0.2563) and Study sites (p = 0.175) for **ITS** and D. plant organs (p = 0.312), **E.** epiphytes and endophytes (p = 0.492), **F.** Study sites (p = 0.726) for 28S rRNA gene in cashew phyllosphere. The significance test was confirmed using TukeyHSD at 95% confidence Interval (CI).

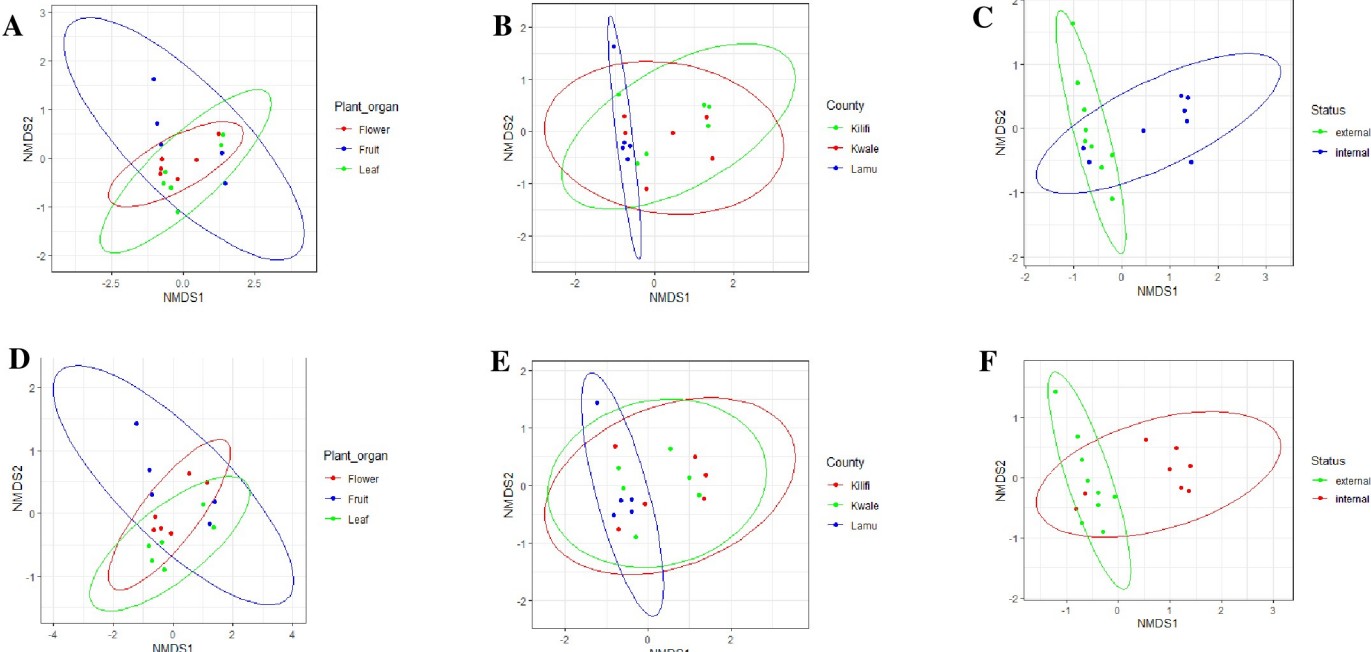

**Fig 3. Non-metric multidimensional scaling (NMDS) showing the results of the beta diversity analysis using the unweighted UniFrac metrics. A**. plant organs, **B.** study sites, **C**. epiphytic and endophytic for ITS, **D**. plant organs, **E**. study sites, **F**. epiphytic and endophytic for 28S rRNA gene in cashew phyllosphere.

using a distance-based redundancy test (db-RDA) based on unweighted UniFrac metrics revealed no significant difference of fungal communities between plant organs and study sites, unlike epiphytes and endophytes for both ITS and 28S rRNA gene which had a significant difference (Fig 3). The significance of the difference among the epiphytes and endophytes was confirmed using the ANOSIM R statistic (R: 0.3273, for 28SrRNA and R: 0.3034 for ITS).

## Fungal composition and abundance of the phyllosphere

Based on the analysis of complete ITS and 28S rRNA gene data sets, ITS had diverse fungal phyla with *Ascomycota* as the most abundant, followed by *Basidiomycota*, unclassified fungi (*Fungi phy Incertae sedis*) and *Glomeromycota* (Fig 4A). In contrast, the 28S rRNA gene dataset showed *Ascomycota* as the most abundant, followed by *Basidiomycota* (Fig 4B). The fungal communities within *Ascomycota* were dominant in all plant organs as epiphytes and endophytes (Fig 4C and 4D). However, phylum *Basidiomycota* in the ITS dataset was abundant among endophytic flowers, followed by unclassified fungi (*Fungi Phy Incertae sedis*) (Fig 4A). Compared to endophytes, the epiphytic community for ITS OTUs was more enriched with a high abundance of class *Saccharomycetes* (Fig 4C). The 28S rRNA gene epiphytic community was highly enriched with class *Saccharomycetes* that were the most abundant followed by *Dothideomycetes* and *Leotiomycetes* (Fig 4D). The class *Saccharomycetes* of ITS sequences was abundant among epiphytic fruit samples, while *Tremellomycetes* abundant in epiphytic leaf and endophytic flower samples (Fig 4C). In terms of fungal orders for ITS OTUs, *Saccharomycetales* and *Tremellales* were the most abundant in epiphytic fruit and endophytic flower samples respectively (S4A Fig in S1 File). Among the 28S rRNA gene, the class *Dothideomycetes* comprised of the orders *Dothideales* and *Capnodiales* in epiphytic and endophytic

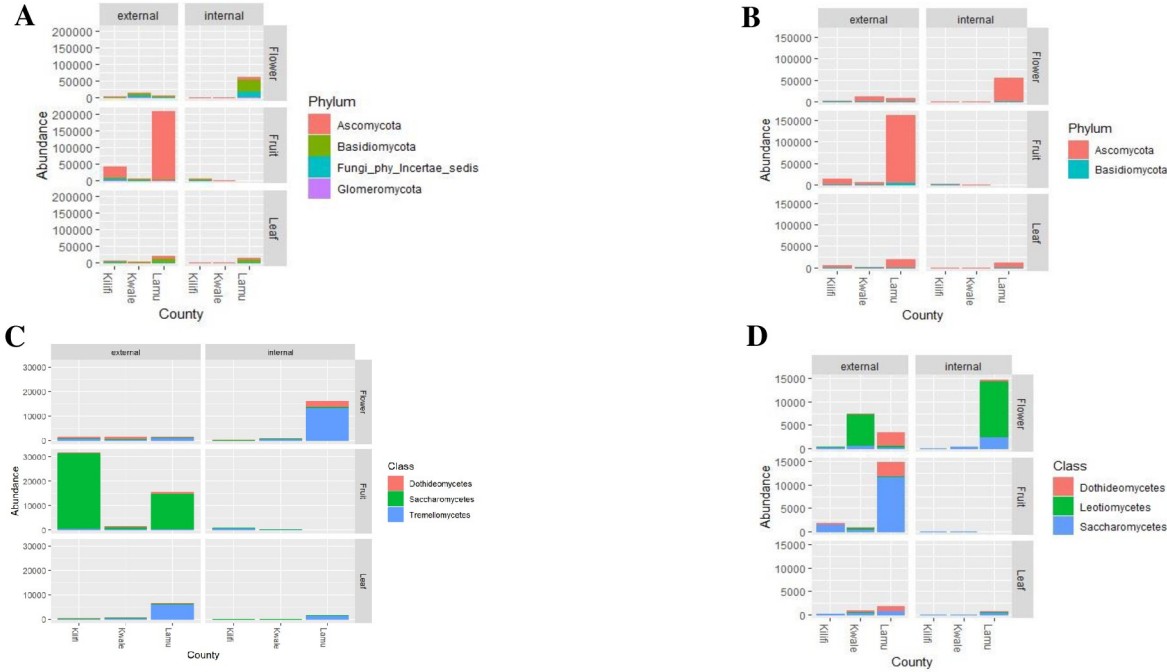

**Fig 4.** Fungal relative abundance and composition among the cashew phyllosphere across the study sites: **A**. Phyla abundance and composition using **ITS** sequence, **B**. Phyla abundance and composition using 28S rRNA gene sequence, **C**. Class abundance and composition using **ITS** sequence, **D.** Class abundance and composition using 28S rRNA gene sequence.

communities (S4B Fig in S1 File). Additionally, the order *Saccharomycetales* was most abundant between epiphytic fruit (S4 Fig in S1 File).

Fungal composition and abundance across the study sites showed a relatively similar distribution for ITS and 28S rRNA OTUs. The fungal communities of the cashew phyllosphere from Kwale had class *Dothideomycetes* and *Saccharomycetes* as most abundant among epiphytic flower and fruits, respectively (S5A Fig in S1 File). In contrast, the 28S rRNA gene OTUs of Kwale cashew phyllosphere had class *Leotiomycetes* as most abundant, followed by *Saccharomycetes* among epiphytic flower samples (S5B Fig in S1 File). The ITS OTUs of cashew phyllosphere from Lamu had *Saccharomycetes* among epiphytic fruit samples and *Tremellomycetes* among endophytic flower samples (Fig 4C). Similarly, the ITS microbial data for cashew phyllosphere from Kilifi constituted *Saccharomycetes* as most prominent among epiphytic fruit samples (Fig 4C). Ecological functions associated with the cashew phyllosphere fungal communities showed more saprotrophs across the fungal classes followed by saprotrophs-symbiotrophs (Fig 5A and 5B). The pathotrophs and pathrotroph-symbiotrophs were most abundant among class *Dothideomycetes* and *Eurotiomycetes* (Fig 5A).

## Fungal community network structure

Fungal community network structure analysis based on a correlation matrix showed a close relationship of nodes among microbial communities (S6 Fig in S1 File). Hierarchical clustering of fungal microbial communities among cashew phyllosphere based on Weighted Unifrac for ITS and 28S rRNA gene revealed a robust genetic structure of species co-existence (S7 Fig in S1 File). The fungal communities within had solid connections and moderate a modular structure. The close connectivity within and among different fungal phyla for both ITS and 28S

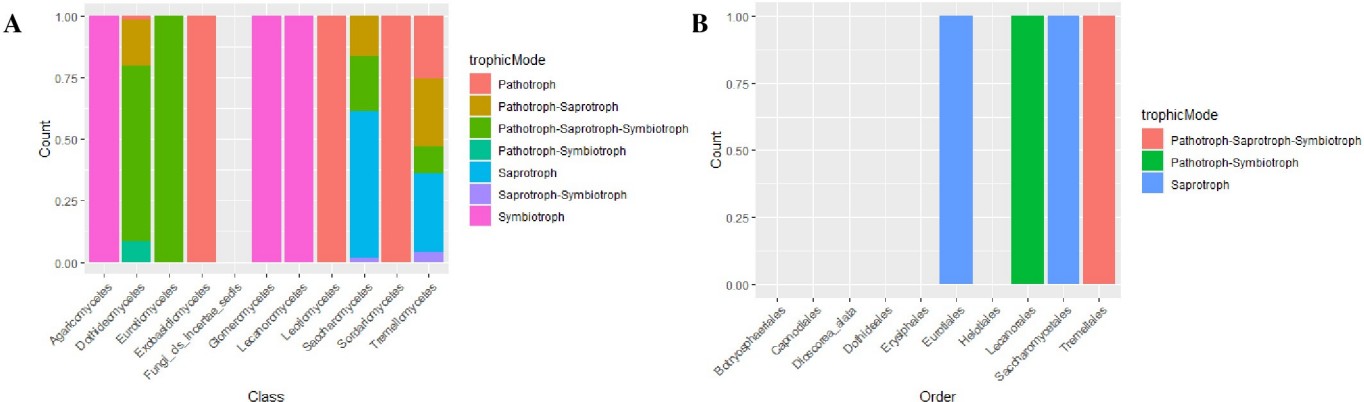

**Fig 5.** Functional prediction analysis of associated Cashew phyllosphere fungal communities: **A**. ITS associated fungal communities, **B**. 28S associated fungal communities.

rRNA gene was visualized and revealed in the modularity values (Fig 6). A network structure analysis within *Basidiomycota* revealed a high abundance and similarity among class *Tremellomycetes* compared to *Exobasidiomycetes* and *Agaricomycetes* (Fig 7A). Within the epiphytes and endophytes fungal communities *Ascomycota* class labels at edges which authenticate a high abundance and similarity of class *Saccharomycetes* compared to *Dothideomycetes* and *Eurotiomycetes* (Fig 7B). The phyla showed dissimilarity to the edge thickness connecting them (Fig 8). Community structure analysis per phylum with class label at the edges binds with taxa abundance and similarity among each class.

## Phylogenetic relationship of the fungal communities

The fungal communities from the ITS (Fig 9) and 28S rRNA gene (Fig 10) in the topology showed significant bootstrap values with high confidence levels among fungal clades within individual phyla. The phylum *Fungi Phy Incertae sedis* (unclassified fungi) and *Basidiomycota* branched off at a significant node value, indicating their evolutionary relationship (Fig 9). *Aurobasidium thailandase* species under *Dothideomycetes* branched off with *Pichia* sp with a

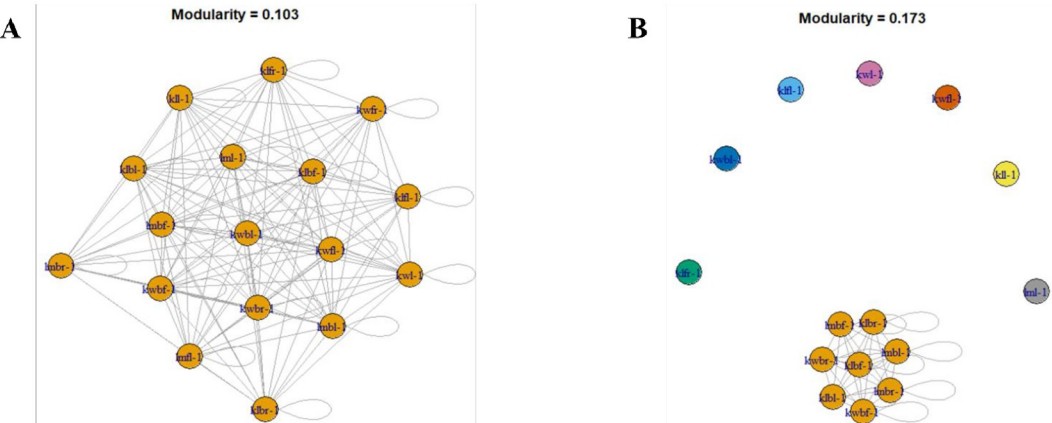

**Fig 6. Clustering of community network structure among different fungal phyla of the cashew phyllosphere for ITS dataset.**

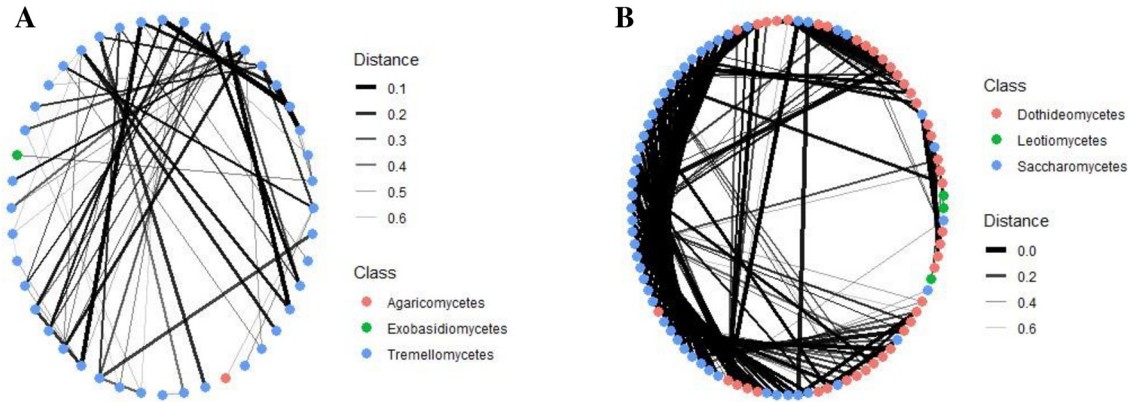

**Fig 7. Clustering of fungal community network structure of the cashew phyllosphere for ITS. A.** *Basidiomycota*, **B.** *Ascomycota*.

significant node value (0.791) (Fig 10). The species in the genus *Clavispora* are identical in simultaneity with their abundance and significant node values of 0.997 between *Clavispora lustinaniale* and *Clavispora* sp (S8 Fig in S1 File). Class *Dothideomycetes*, constituting of *Lasiodiplodia pseudotheobromae* and *Dothideomycetes* sp of order *Botryosphaeriales* had a significant node value of 0.936 (S9 Fig in S1 File). *Scleroconidioma* sp and *Pseudosydowia* sp fungal species of *Dothideales* order showed close relatedness with a node value of 0.858 (S9 Fig in S1 File). Fungal orders *Erysiphales*, *Eurotiales*, *Dothideales*, *Saccharamycetales Helotiales* and *Capnodiales* of *Ascomycota* for 28S rRNA gene clustered with significant node value 0.945 (Fig 11). The fungal order of phyla *Ascomycota* and *Basidiomycota* clustered together and had a robust node value above 0.700. The species within the fungal orders *Agaricales*, *Athelliales*, *Tremellales* and *Trichosporonales* had a significant node value (0.997) (Fig 12).

## Discussion

Fungi communities inhabit different plant surfaces such as the phyllopane, carposphere asthenosphere, and internal tissues (endosphere), which are pivotal for the development and

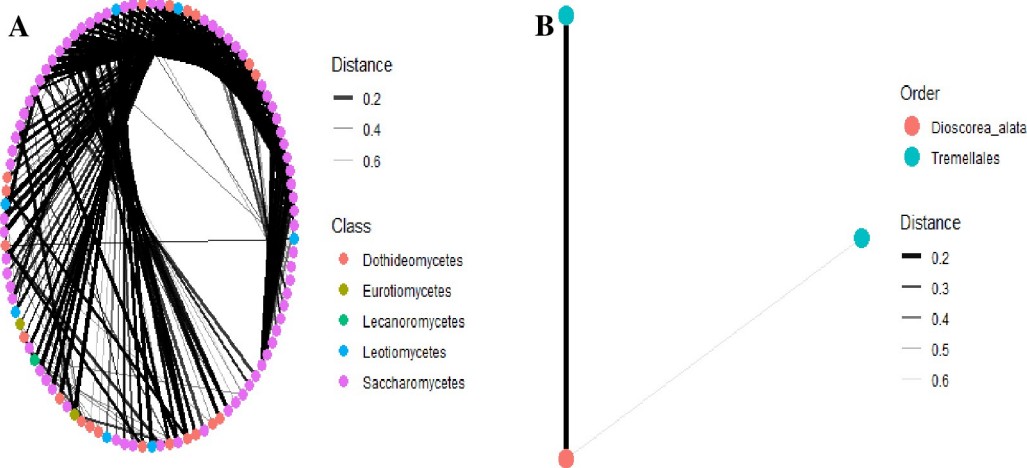

**Fig 8. Clustering of fungal community network structure of the cashew phyllosphere for 28S rRNA gene. A.** *Ascomycota*, **B.** *Basidiomycota*.

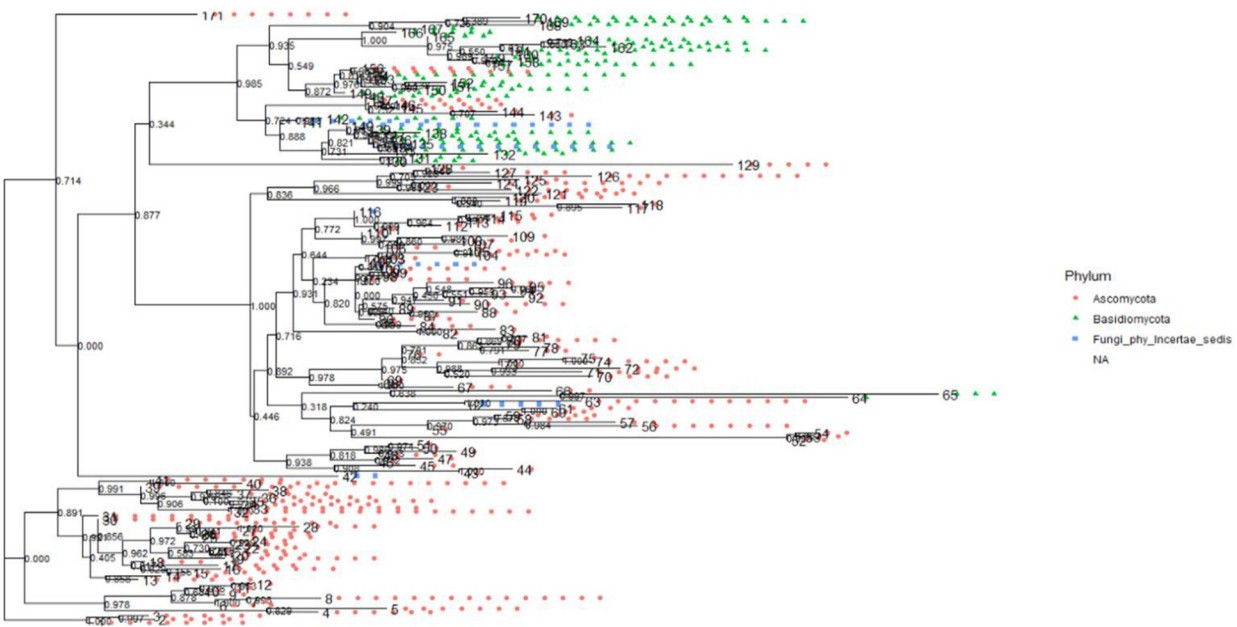

**Fig 9. Evolutionary relationship between phylum *Ascomycota*, *Basidiomycota* and *Fungi Phy Incertate sedis* (unclassified fungi) in Kwale, Kilifi and Lamu for ITS dataset.** The phylogenetic tree was inferred from the maximum likelihood.

protection of plants in response to biotic and abiotic stressors [6]. Soil properties, wind composition, air and water [32], as well as different plant compartments [33] influence the fungal communities. We investigated the fungal communities from different plant organs as epiphytes and endophytes of cashew phyllosphere across our study sites. Epiphytic and

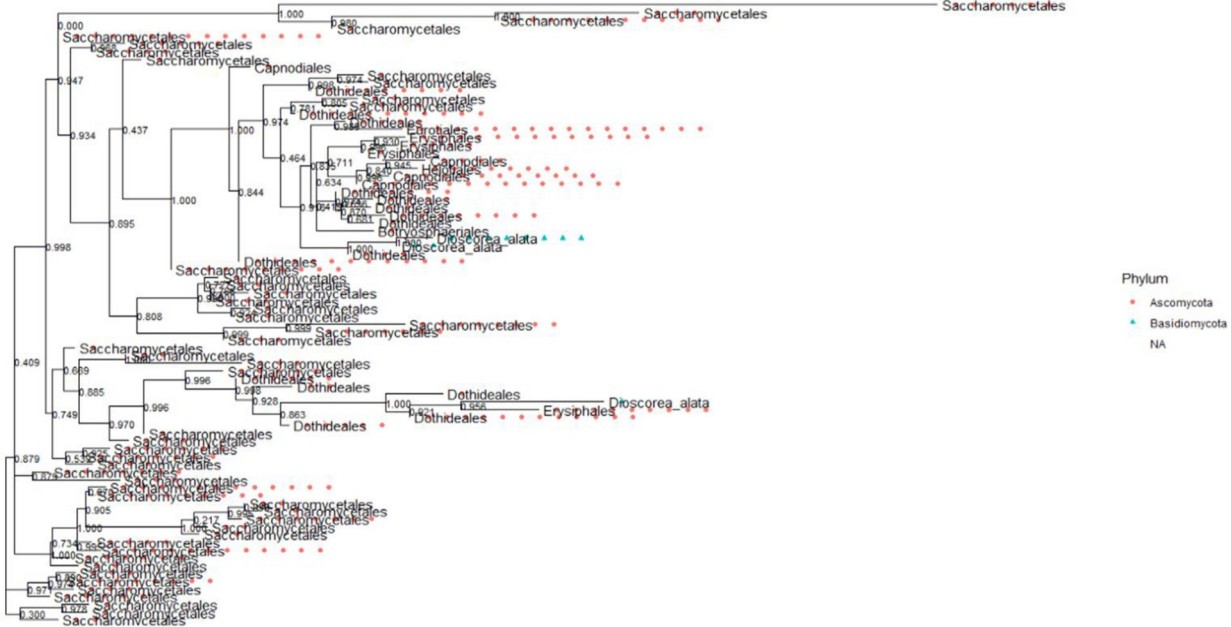

**Fig 10. Evolutionary relationship among phylum *Ascomycota* in Kwale, Kilifi and Lamu Lamu for 28S rRNA gene dataset.** The phylogenetic tree was inferred from the maximum likelihood.

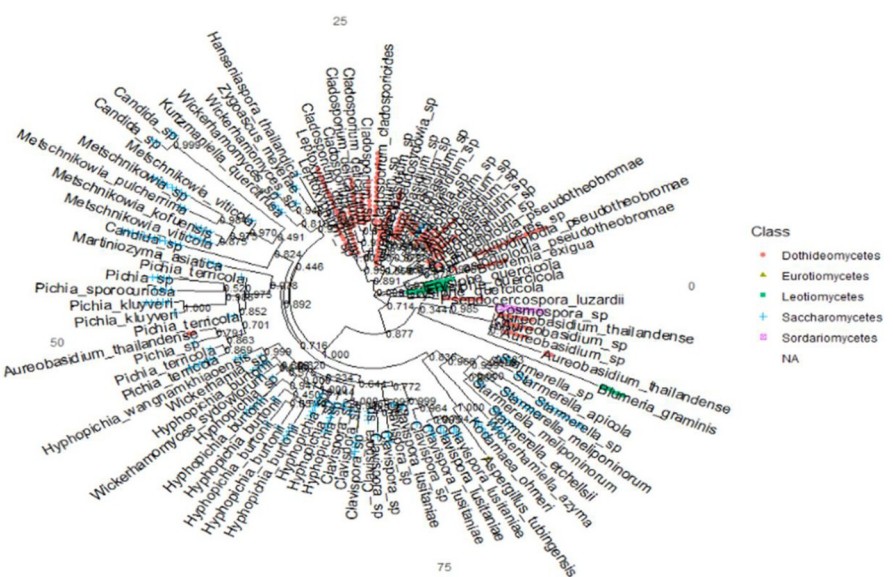

**Fig 11. Evolutionary relationship among phylum _Ascomycota_ and _Basidiomycota_ in Kwale, Kilifi and Lamu for 28S rRNA gene.** The phylogenetic tree was inferred using the maximum likelihood.

endophytic microbiomes harbour specific fungal communities with associated ecological functions as symbiotrophs, saprotrophs and pathogens [34]. The fungal epiphytes adapt to external conditions, unlike endophytes, due to their ecological roles as decomposers and secreting antimicrobial metabolites [35]. The alpha and beta diversity of the fungal communities within the different plant organs and study sites was found to be insignificant. Our results contrasts with other metagenomics studies on strawberries [11] and olive trees [36] which revealed a significant difference in composition and diversity among the phyllosphere. In their work, the different plant organs (leaf, flower, and mature and immature fruit) significantly differed, showing diverse fungal community among leaves compared to the fruit. The differences observed by [36] was due to a higher surface/volume ratio of leaves compared to fruits. Despite the

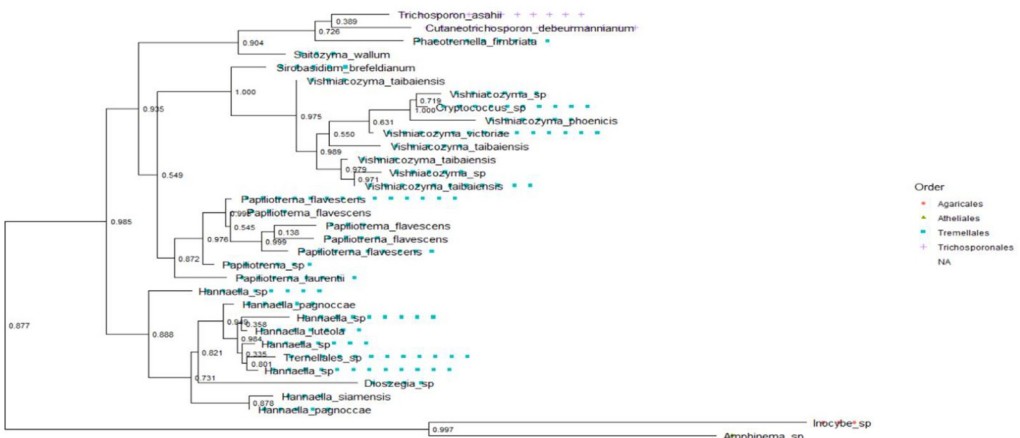

**Fig 12. Evolutionary relationship among phylum _Basidiomycota_ in Kwale, Kilifi and Lamu for ITS.** The phylogenetic tree was inferred from the maximum likelihood.

morphological differences among cashew leaves and fruits, their microbial composition and diversity were non-significant. We therefore hypothesize that the collection of samples from open fields (uncontrolled condition) and different cashew varieties that were not considered could have had an influence. Despite the non-significant fungal composition and diversity among the cashew leaves and fruits, the phytochemical profiles such as sugars, volatile compounds and secondary metabolites between leaves and fruits influenced the fungal richness and evenness. The agro-ecological zones of the coastal lowlands 4 (Cl4) share similar temperature and humidity profiles [37]. This may play a role in influencing the community composition and diversity.

The diversity of epiphytes and endophytes showed a significant difference between the two microenvironments. Epiphytes are influenced by extreme external conditions such as wind speed, temperature, relative humidity, and solar radiation, whereas endophytes are majorly influenced by the host plants' immune systems and nutrient contents within internal tissues [10]. Our findings are similar to those of [6, 9] who reported distinct fungal epiphytic and endophytic communities among the Mediterranean and mangrove ecosystem. The fungal endophytes typically penetrate the plant tissues through a horizontal transmission mechanism, as observed in woody plants [6].

The fungal community associated with the cashew phyllosphere in the current study was diverse comprising of phyla *Ascomycota*, *Basidiomycota*, *Fungi phy Incertae sedis* (unclassified fungi) and *Glomeromycota*. The metabarcoding approach previously used by [11] reported a rich fungal community (316 OTUs) associated with leaves, flowers and fruits of strawberry plants growing in Lamezia Terme, Italy. Their findings corroborate our results of OTU abundance and phylum composition, thus authenticating the reproducibility of amplicon sequencing approach in microbiome studies. Here, we postulate that the diverse fungal phyla detected, including the unclassified fungi (*Fungi phy Incertae sedis*) among the epiphytes and endophytes of cashew phyllosphere could be associated with diverse environmental variations and microenvironmental differences.

The epiphytic cashew fungal community was evenly richer and more abundant compared to the endophytic community. Similar findings have been described for the phyllosphere of other woody plant ecosystems such as in *Camellia japonica* [38] and *Coffea arabica* [39]. The composition of the epiphytic fungal community in cashew phyllosphere was entirely different from that of endophytes, with class *Saccharomycetes* being the most abundant in epiphytic fruits and *Tremellomycetes* in endophytic flowers. Similar findings of distinct fungal assemblages among epiphytes and endophytes were reported by [10] who reported classes *Saccharomycetes*, *Dothideomycetes* and *Eurotiomycetes* as the most abundant among tomato plants grown in greenhouse. Moreover, other researchers [40] reported a difference in epiphytic and endophytic communities with 478 fungal isolates, where 279 were epiphytic and only 199 were endophytic among the leaves of *Eucalyptus citriodora* Hook. Epiphytic communities were enriched by some fungal species with melanised hyphae/spores belonging to class *Dothideomycetes* and *Tremellomycetes* [40]. Classes *Saccharomycetes*, *Dothideomycetes* and *Tremellomycetes* were the most abundant among the epiphytic fruit samples compared to the endophytes (Fig 4). The cashew leaves are covered with glandular trichomes for secretion of sugar esters on plant leaf surfaces [41]. The sugar exudates might provide a selective source of nutrition for the growth of *Saccharomycetales* and *Capnodiales*. This is in conjunction with the different physiological roles and nutritional profiles of cashew leaves and flowers besides having the same exposure to environmental parameters.

Class *Tremellomycetes* of phylum *Basidiomycota* was highly enriched in endophytic flowers and among epiphytic samples. Similar findings were reported by [42] in poplar ecosystem and [9] among mangroves. *Tremellomycetes* encompass vital ecosystem processes promoting

nutrient cycling and breaking down of organic matter enhancing their interactions with other fungal communities [43]. Classes *Saccharomycetes* and *Dothideomycetes* in epiphytic fruit samples underpins their pivotal role in microbial interaction towards ecosystem functioning. Our findings are similar with the work of [10] which identified *Saccharomycetales* in the pericarp and the jelly around seeds of tomatoes. The abundance of *Saccharomycetes* and *Dothideomycetes* among the fruit samples corresponds to their role as phytopathogens and saprophytes in cashew phyllosphere ecosystem. *Pichia terricola*, *Candida* sp. and *Clavispora* sp. of class *Saccharomyces* are known to enhance hydrocarbon degradation, bioremediation and insect interactions [44, 45]. Hence, their presence among the cashew phyllosphere resonates with their roles. Yeast species of class *Saccharomycetes* are key agents for plant growth and development through nutrient cycling and decomposition of organic matter [45]. *Issatchenkia orientalis* is a known yeast species in biotechnology where it is used in industrial applications for enzyme and recombinant protein production [46, 47]. These yeast species influence fruiting during pollination and flowering in *Anacardiceae* plant species and consequently promote cashew plant productivity and yield [48].

Fungal communities among the study sites consisted of class *Dothideomycetes*, *Saccharomycetes Leotiomycetes*, *Tremellomycetes* and *Exobasidiomycetes*. *Lasiodiplodia pseudotheobromae*, *Alternaria* sp. and *Cladosporium cladosporioides* are notable phytopathogens of class *Dothideomycetes*. The species interaction between class *Dothideomycetes* and those in *Tremellomycetes* presented a unique ecological interaction which may promote cashew growth and development. These unique interspecies interactions among the cashew phyllosphere opens a research gap to understand the cashew-fungi interaction. Class *Leotiomycetes* was enriched among epiphytic flowers, which contrasts with the results of [11] on strawberry and [49] on pears. They who both reported *Leotiomycetes* enrichment among fruits samples. *Exobasidiomycetes* (smut fungi) are obligate phytopathogens which colonizes the plant surfaces [43] and have a unique life cycle (dikaryotic division). This phytopathogen was reported among tomato plants within greenhouse settings [10]. *Exobasidiomycete* presence in cashew confirm its wide range of hosts. The predicted function among the fungal communities of cashew phyllosphere reveals a useful ecological function as saprotrophs and pathotrophs.

Disentangling cashew fungal interactions by ecological network analysis is key to understanding the mechanisms of species co-existence and microbial load in an ecosystem [9]. The fungal community network structure among cashew phyllosphere comprising both epiphytes and endophytes presents a stable structure with significant similarity within the fungal communities. The epiphytic and endophytic fungal community network structures show stronger connections within each phylum. Phyla *Ascomycota*, *Basidiomycota*, *Glomeromycota* and the unclassified fungi (*Fungi phy Incertae sedis*) among the epiphytes and endophytes are strongly connected within themselves. Our results verify those of [9] on different mangrove plants, reflecting on the robustness and resilience of the cashew microbiome against perturbations.

The phylogenetic relationship among different fungal communities, both between and within the phylum, infers to a conservative evolution with robust genetic relatedness. The fungal communities of *Ascomycota*, *Basidiomycota* and *Fungi phy Incertae sedis* clustered together, signifying high conversation of the ITS and 28S rRNA gene of the fungal genome [50]. The *Fungi phy Incertae sedis* clustered among *Basidiomycota* with a statistically significant node value [51]. The phylogenetic relationship among the different fungal communities using the ITS revealed a similar finding by [11] who reported sequence types for each fungal community clustering together with 1000 bootstrap replications, underscoring the conservatory nature of the ITS barcode. Classes *Saccharomycetes*, *Dothideomycetes*, *Sordariomycetes*, *Eurotiomycetes* and *Leotiomycetes* of phylum *Ascomycota* authenticate the robust evolutionary stability of the taxonomic molecular makers used in this study. Fungal species of *Basidiomycota* clustered

together with a close genetic relatedness. This level of conservative evolution strongly relates to genetic alteration and diversification occurring over limited time [52, 53]. The fungal communities among the cashew phyllosphere in our study point to the irrefutable knowledge of fungal diversity and evolution. Our findings are noteworthy and relevant for more research insights towards enhanced understanding of cashew fungal abundance and diversity.

## Conclusion

In conclusion, results of the present study provide a comprehensive picture of the fungal diversity in the cashew phyllosphere. Majority of the detected sequences were identified up to species level making it possible to support assumptions regarding their ecological role on the cashew leaf, flower and fruit. Collectively, this study reinforces the importance of investigating fungal biodiversity among the cashew phyllospehere and highlighting the need for more detailed analyses to understand the key drivers of fungal abundance and composition. These results will improve the understanding of the community structure and dynamics of the phyllosphere microorganisms in cashew plants because they might be important in supporting cashew tree survival in Kenya.

## Supporting information

**S1 File.**
(PPTX)

**S2 File.**
(DOCX)

**S3 File.**
(DOCX)

## Acknowledgments

The authors are grateful to Pwani University for providing the laboratory space at the Pwani University Biosciences Research Centre (PUBReC) to perform this work.

## Author Contributions

**Conceptualization:** Wilton Mwema Mbinda.

**Formal analysis:** Dennis Wamalabe Mukhebi, Colletah Rhoda Musangi.

**Funding acquisition:** Everlyne Moraa Isoe, Wilton Mwema Mbinda.

**Investigation:** Dennis Wamalabe Mukhebi, Colletah Rhoda Musangi, Wilton Mwema Mbinda.

**Methodology:** Dennis Wamalabe Mukhebi, Colletah Rhoda Musangi, Wilton Mwema Mbinda.

**Project administration:** Wilton Mwema Mbinda.

**Resources:** Wilton Mwema Mbinda.

**Supervision:** Everlyne Moraa Isoe, Johnstone Omukhulu Neondo, Wilton Mwema Mbinda.

**Validation:** Wilton Mwema Mbinda.

**Writing – original draft:** Dennis Wamalabe Mukhebi, Colletah Rhoda Musangi, Wilton Mwema Mbinda.

**Writing – review & editing:** Everlyne Moraa Isoe, Johnstone Omukhulu Neondo, Wilton Mwema Mbinda.

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
