## [Decision Letter · Decision Letter 0]

17 May 2024

PONE-D-24-16486Endophytic and epiphytic metabarcoding reveals fungal communities on cashew phyllosphere in KenyaPLOS ONE

Dear Dr. Mbinda,

Thank you for submitting your manuscript to PLOS ONE. After careful consideration, we feel that it has merit but does not fully meet PLOS ONE’s publication criteria as it currently stands. Therefore, we invite you to submit a revised version of the manuscript that addresses the points raised during the review process.

We look forward to receiving your revised manuscript.

Kind regards,

Eugenio Llorens

Academic Editor

PLOS ONE

Journal Requirements:

"This research was supported by the National Research Fund, Kenya (NRF/2/MMC/158) and The World Academy of Sciences (RGA No. 21-302 RG/BIO/AF/AC_G). "

Reviewers' comments:

Reviewer's Responses to Questions

**Comments to the Author**

1. Is the manuscript technically sound, and do the data support the conclusions?

Reviewer #1: Yes

Reviewer #2: Yes

2. Has the statistical analysis been performed appropriately and rigorously? 

Reviewer #1: I Don't Know

Reviewer #2: Yes

3. Have the authors made all data underlying the findings in their manuscript fully available?

Reviewer #1: Yes

Reviewer #2: Yes

4. Is the manuscript presented in an intelligible fashion and written in standard English?

Reviewer #1: Yes

Reviewer #2: No

5. Review Comments to the Author

Reviewer #1: The manuscript titled " Endophytic and epiphytic metabarcoding reveals fungal communities on cashew phyllosphere in Kenya" is comprehensive and interesting, especially contribute to the understanding of the microbiome of cashew. Here are the comments:

1. Please provide details in the M&M section, such as the replications of each tissue sample; they should be sufficiently supplied.

2. What’s the portion of your DNA regarding microbes and plants? How to deal with this in this study?

3. Could the term “phyllosphere” cover your study about different plant organs?

4. The authors use both ITS and 28S rRNA to barcode the presence of the fungi in and on the plants. Why you use this approach? What’s the pros and cons? How to interpret the difference? It is better to further discussed in this article

Other suggestions:

5. L130: The term “Isolation” should be replaced since the microbes were not isolated and cultured.

6. L137-143: How large are the samples? How many samples? Please provide enough details in the M&M section.

7. L154: It seems like “county”, not “country”.

8. L155: NL4, instead of NL

9. L202: b ITS?

10. L200: OTSs

11. L213: 28S rRNA

12. L218, L222: phy Incertae sedis or Phy Incertae sedis? Make sure they are the same, check the entire article.

13. L224, L225: Saccharomycetes, italicized or not? Make sure they are the same, check the entire article.

14. L273: sp. Check the entire article.

15. L277: sp. Should not be italicized.

16. L281: Saccharamycetales, Helotiales

17. L366: “Candid sp” is incorrect.

18. Make sure the reference style follow the Submission Guidelines. Some of the writing is inconsistent.

19. Legend of Supplementary 7 is not consistent, check the entire article.

Reviewer #2: The research work presented in this manuscript is rigorously designed, with experiments conducted meticulously and data analysis performed appropriately. However, I would like to offer some constructive feedback regarding the clarity and accessibility of the writing style.

It is advisable for the authors to aim for precision in their writing and adopt a more accessible language, particularly in the Abstract, Introduction, Discussion, and Conclusion sections. This would make the content more understandable to a broader audience, minimizing the use of excessive technical jargon.

For instance, the Introduction should concentrate more on the issues related to cashew cultivation and the potential roles of microbiota. It should also discuss previously isolated endophytes from this or similar plant species, setting the stage for proposing potential solutions. Typically, an effective introduction outlines a problem, proposes a solution, and previews the methods used to explore that solution.

In the Conclusion, the key takeaways are somewhat vague concerning the specific causes of problems in cashew trees, such as pathogens. I recommend revising parts of the manuscript to enhance clarity and digestibility for the general reader.

A more direct approach would be to clearly state that cashew production faces significant challenges, primarily due to fungal pathogens and the overuse of agricultural land. The manuscript should then detail how the conducted meta-analysis addresses these challenges, presenting findings such as the structures of endophytic and epiphytic mycobiome communities and subsequently proposing practical solutions for the identified issues in cashew cultivation.

6. PLOS authors have the option to publish the peer review history of their article (what does this mean?). If published, this will include your full peer review and any attached files.

Reviewer #1: **Yes: **Yuan-Min Shen

Reviewer #2: **Yes: **Mahmoud W. Yaish

---

## [Author Response · Author response to Decision Letter 0]

27 May 2024

Responses to reviewers' comments 

Reviewer Comment Response 

Reviewer #1 

 Please provide details in the M&M section, such as the replications of each tissue sample; they should be sufficiently supplied. Tissue samples (Leaf, flower and fruit) were collected across the three study sites. DNA was acquired from each sample in duplicate. However, concerning our study objectives that was anchored on creating insights into the abundance and composition of available, we pooled the DNA together creating a representative of 18 samples. In the manuscript, the sample collection per county has been indicated and the grand total stated.

 What is the portion of your DNA regarding microbes and plants? How to deal with this in this study? The microbes DNA (fungal gDNA) was of interest. Concerning the study goals, the use of universal fungal barcodes was a priority, therefore we used primers targeting the ITS and 28S rRNA for fungal DNA therefore excluding the plant DNA. 

 Could the term “phyllosphere” cover your study about different plant organs? Phyllosphere generally refers to all above surface plant organs and therefore in using this word I have specified the key areas of interest in this work, these are (Leaf, flower and fruit).

 The authors use both ITS and 28S rRNA to barcode the presence of the fungi in and on the plants. Why you use this approach? What is the pros and cons? How to interpret the difference? It is better to further discussed in this article The ITS and 28S rRNA microbe detection vary in rational to abundance and composition because they are in different databases which are updated differently. In addition 28S rRNA database is very recent therefore it is less enriched than ITS however it has notable microbes that are not present in ITS as revealed in this work. This approach is ideal in maximising on detection of available microbial communities that is insightful by creating a plethora of information about microbial communities present. However, the cost incurred tend to limit in addition to sometimes having similar composition although the abundance will always differ. This is important to have the entire databases enriched and updated so the future studies.

 Other suggestions The other suggestions made have been acted upon and the coherence of content flow established

Reviewer #2 Address Abstract, Introduction, Discussion and Conclusion sections In response to the feedback provided, the key sections have been addressed as directed and the changes are trackable in the main manuscript having track changes. Additionally, the need for clarity and precision of the work therein the manuscript has been adhered.

---

## [Decision Letter · Decision Letter 1]

4 Jun 2024

Endophytic and epiphytic metabarcoding reveals fungal communities on cashew phyllosphere in Kenya

PONE-D-24-16486R1

Dear Dr. Mbinda,

We’re pleased to inform you that your manuscript has been judged scientifically suitable for publication and will be formally accepted for publication once it meets all outstanding technical requirements.

Kind regards,

Eugenio Llorens

Academic Editor

PLOS ONE

Additional Editor Comments (optional):

Reviewers' comments:

Reviewer's Responses to Questions

**Comments to the Author**

1. If the authors have adequately addressed your comments raised in a previous round of review and you feel that this manuscript is now acceptable for publication, you may indicate that here to bypass the “Comments to the Author” section, enter your conflict of interest statement in the “Confidential to Editor” section, and submit your "Accept" recommendation.

Reviewer #1: All comments have been addressed

2. Is the manuscript technically sound, and do the data support the conclusions?

Reviewer #1: Yes

3. Has the statistical analysis been performed appropriately and rigorously? 

Reviewer #1: N/A

4. Have the authors made all data underlying the findings in their manuscript fully available?

Reviewer #1: Yes

5. Is the manuscript presented in an intelligible fashion and written in standard English?

Reviewer #1: Yes

6. Review Comments to the Author

Reviewer #1: (No Response)

7. PLOS authors have the option to publish the peer review history of their article (what does this mean?). If published, this will include your full peer review and any attached files.

Reviewer #1: **Yes: **Yuan-Min Shen

---

## [Editor Report · Acceptance letter]

13 Jun 2024

PONE-D-24-16486R1 

PLOS ONE

Dear Dr. Mbinda, 

I'm pleased to inform you that your manuscript has been deemed suitable for publication in PLOS ONE. Congratulations! Your manuscript is now being handed over to our production team.

Kind regards, 

on behalf of

Dr. Eugenio Llorens 

Academic Editor

PLOS ONE